Brief Communication

# RS-FISH: precise, interactive, fast, and scalable FISH spot detection

Ella Bahry[1,7], Laura Breimann [1,2,7], Marwan Zouinkhi[1,3,7], Leo Epstein[1,4], Klim Kolyvanov[1], Nicholas Mamrak[5], Benjamin King[5], Xi Long[3], Kyle I. S. Harrington [1,4]✉, Timothée Lionnet [5,6]✉ & Stephan Preibisch [3]✉

Fluorescent in-situ hybridization (FISH)-based methods extract spatially resolved genetic and epigenetic information from biological samples by detecting fluorescent spots in microscopy images, an often challenging task. We present Radial Symmetry-FISH (RS-FISH), an accurate, fast, and user-friendly software for spot detection in two- and three-dimensional images. RS-FISH offers interactive parameter tuning and readily scales to large datasets and image volumes of cleared or expanded samples using distributed processing on workstations, clusters, or the cloud. RS-FISH maintains high detection accuracy and low localization error across a wide range of signal-to-noise ratios, a key feature for single-molecule FISH, spatial transcriptomics, or spatial genomics applications.

New FISH-based imaging methods are continuously being developed to gain insights into cellular processes, for example, by resolving the subcellular localization of single RNA molecules[1,2] or subnuclear 3D arrangement of DNA regions[3,4]. Classically, single-molecule FISH (smFISH) has been used to visualize individual mRNA molecules for single genes in small samples[1,2]. New methods that employ probe amplification, probe multiplexing, or barcodes are driving the fields of spatial transcriptomics and spatial genomics, enabling the subcellular visualization of thousands of genes with single-molecule sensitivity in complex tissues[5–10], as well as entire chromosomes with high resolution at nanometer scale[3].

Extracting information from smFISH, spatial transcriptomics, or spatial genomics images relies on the precise detection of diffraction-limited spots. Important properties of spot-detection software include accuracy and speed of detection, as well as being accessible to researchers. Recently, scalability to large datasets has become important because the detection of subtle transcriptional changes relies on the analysis of thousands of smFISH images[11,12], increasingly large samples in the tera-byte range are being imaged[13], and spatial-transcriptomics methods are being applied to increasingly

large samples, with many rounds of sequential hybridization and imaging (Fig. 1 and Supplementary Notes). Several methods are available; however, all commonly used packages do not allow interactive parameter tuning, which makes their application tedious. They also do not scale to large datasets because they are missing out-of-core processing capabilities for large images, have no straightforward path to automation and distribution for large sets of smaller images, and have increased runtimes because of their slower processing times[1,14–18]. To overcome these restrictions, we developed RS-FISH, which uses an extension of Radial Symmetry[19] (RS) to robustly and quickly identify single-molecule spots in 3D with high precision (Fig. 1a). RS-FISH can be run as an interactive, scriptable Fiji plugin[20], as a command-line tool, and as a cluster and cloud-distributable package for large volumes or for datasets consisting of thousands of images (Fig. 1g,h).

RS is an efficient, non-iterative alternative to accurate point localization using Gaussian fitting that was developed for localizing 2D circular objects by computing the intersection point of image gradients (Fig. 1a)[19]. We first derived a 3D version of the RS method, similar to the work of Liu et al.[21] (Methods), that additionally extends to higher dimensions, which has potential for spatiotemporal localization of blinking

[1]Max-Delbrück-Center for Molecular Medicine in the Helmholtz Association (MDC), Berlin Institute for Medical Systems Biology (BIMSB), Berlin, Germany. [2]Department of Biology, Center for Genomics and Systems Biology, New York University, New York, NY, USA. [3]Janelia Research Campus, Howard Hughes Medical Institute, Ashburn, VA, USA. [4]Helmholtz Imaging Platform, Max Delbrück Center for Molecular Medicine, Berlin, Germany. [5]Institute for Systems Genetics and Department of Cell Biology, NYU School of Medicine, New York, NY, USA. [6]Department of Bioengineering, NYU Tandon School of Engineering, Brooklyn, NY, USA. [7]These authors contributed equally: Ella Bahry, Laura Breimann, Marwan Zouinkhi ✉e-mail: kyle.i.s.harrington@gmail.com; Timothee.Lionnet@nyulangone.org; preibischs@janelia.hhmi.org

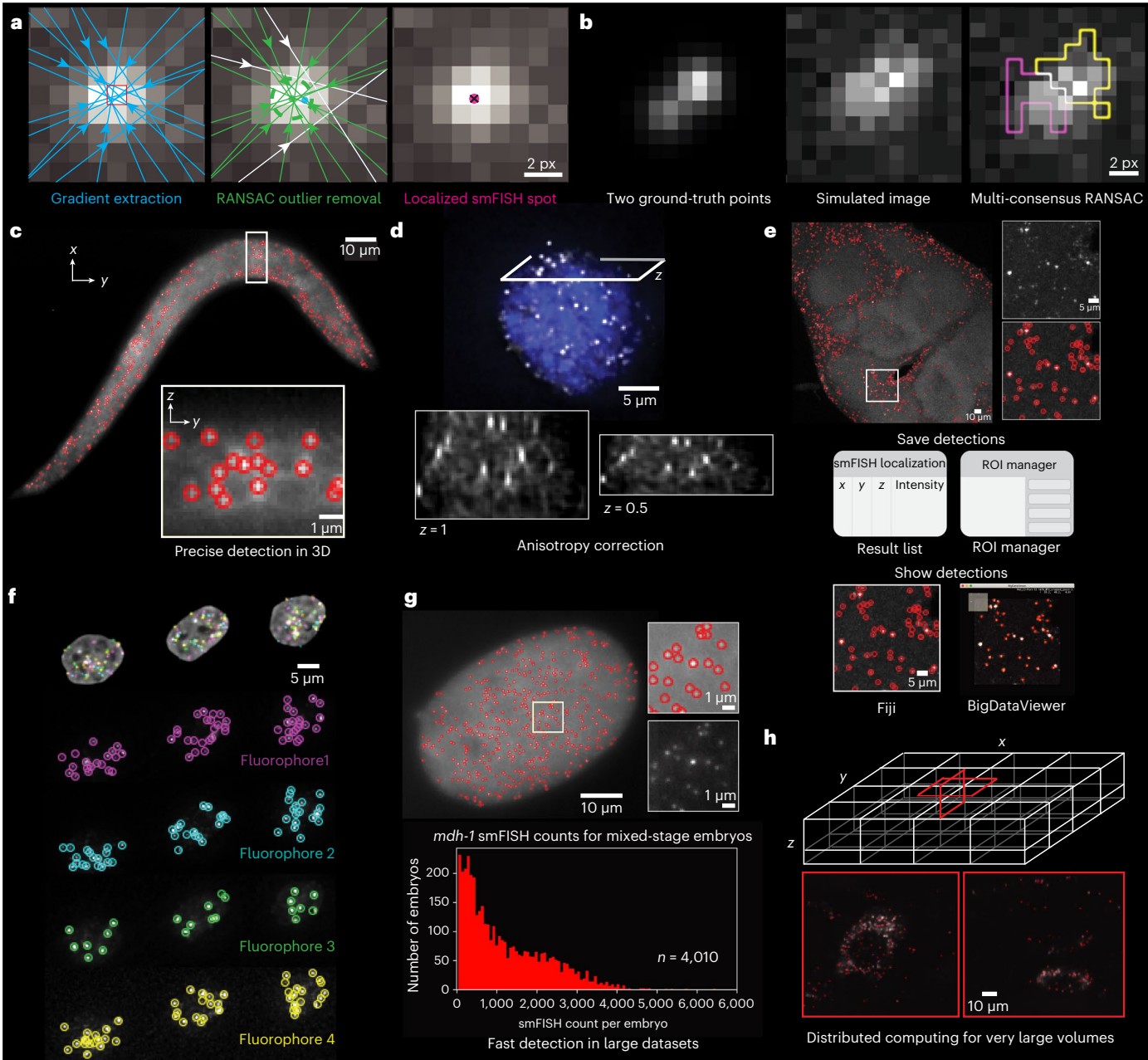

**Fig. 1 | RS-FISH accurately detects fluorescent spots. a**, Illustration depicting single fluorescent spot detection using RS-based RANSAC. Left, gradients (blue lines) calculated in a local patch around a DoG-detected location (red square) for RS fitting. Middle, intensity gradients that agree on a common center point (green gradients, green dot) given a defined error (green dotted circle) are identified using RANSAC outlier removal, and rejected gradients are plotted in white. Using all gradients would lead to a different center point (blue). Right, the final RS-FISH center spot (pink dot with black cross) is computed by intersecting all green (inlier) gradients. **b**, Detecting two close spots using multi-consensus RANSAC. Both points are detected as a single DoG spot owing to a high noise level. Multi-consensus RANSAC identified two independent spots visualized as yellow and pink sets of pixels (that is, gradients). **c**, Single *z* slice through the 3D image of a *C. elegans* larva expressing *lea-1* mRNA (smFISH labeling). Red circles highlight the RS-FISH-detected spots, and the encircled area is shown

as *x*-slice below. Images are representative of four experimental replicates. **d**, To correctly detect spots in anisotropic images, a global scale factor estimated from the data is computed. The example image shows a mouse embryonic stem cell labeled by smFISH for *Cdx2* mRNA. **e**, RS-FISH detections can be exported as result table or CSV file or transferred to the ROI manager, and can be overlaid onto data for inspection using Fiji or BigDataViewer. The example image shows a max projection of five *z*-slices of a *Drosophila* brain with smFISH labeled for *Pura* mRNA[27]. **f**, OligoFISSEQ-labeled[3] PGP1f cells using barcodes with four different fluorophores showing one round of labeling. RS-FISH detected spots are labeled in four different colors. (Images by Nguyen et al.[3]). **g**, RS-FISH scales to large datasets, shown for 4,010 mixed-stage *C. elegans* embryos with *mdh-1* mRNA smFISH labeling. **h**, Large N5 image volumes, like the EASI-FISH 148-GB lightsheet image of a tissue section of the lateral hypothalamus (data by Wang et al.[13]), were analyzed with the Apache Spark version of RS-FISH.

3D spots. Second, we extended RS to support axis-aligned, ellipsoid objects without the need for scaling the image[21], enabling RS-FISH to account for typical anisotropy in 3D microscopy datasets that results from different pixel sizes and point spread functions in the lateral (*x*,*y*)

compared with the axial (*z*) dimensions (Fig. 1d and Methods). Third, the computation speed of RS allowed us to combine RS with robust outlier removal using random sample consensus[22] (RS-RANSAC) to identify sets of image gradients that support the same ellipsoid object given a

specific error for the gradient intersection point (Methods). This allows RS-FISH to identify sets of pixels that support a user-defined localization error for individual spots (Fig. 1a), discriminate close detections (Fig. 1b and Supplementary Fig. SN7.1), and ignore outlier pixels that disturb localization (for example, dead or hot camera pixels).

RS-FISH first generates a set of seed points by thresholding the difference-of-gaussian (DoG)[23] filtered image to identify potential locations of diffraction-limited spots, whose parameters need to be adjusted to the average size (sigma) and intensity (threshold) of the spots. Next, image gradients are extracted from local pixel patches around each spot, which are optionally corrected for non-uniform fluorescence backgrounds. Before RS localization, gradients are rescaled along the axial dimension to correct for dataset anisotropy using an anisotropy factor that depends on pixel spacing, resolution, and point spread function. The anisotropy factor can be computed from the microscopy image itself and does not change as long as acquisition parameters are held constant (Fig. 1d and Methods). Optionally, RS-RANSAC can be run in multi-consensus mode, which performs additional rounds of RANSAC filtering in order to distinguish spots that were too close to one another for the DoG detector to separate them during seed point generation (Fig. 1b, Supplementary Figs. SN7.1 and SN7.2, and Methods). Finally, to avoid potentially redundant detections, spots are, by default, filtered to be at least 0.5 pixels apart from each other. Each spot's associated intensity value is, by default, computed using linear interpolation at the spot's sub-pixel location or can be refined by fitting a Gaussian to the subset of pixels that support the spot as identified by RS-RANSAC.

RS-FISH pixel operations are implemented in ImgLib2 (ref. [24]), and RS fitting and RS-RANSAC are implemented using the image transformation framework mpicbg[25]. All operations can be executed in blocks allowing straightforward parallelization and compute effort scales linearly with the size of the data up to the petabyte range (Methods). Importantly, RS-FISH's parameters can be interactively tuned on small and large datasets using the Fiji plugin (Supplementary Fig. SN8.1). Once the right set of parameters is identified on a representative example image, RS-FISH can be run and macro-scripted in Fiji, or can be executed in a scriptable mode for straightforward parallel execution on compute clusters or cloud services (for example, Amazon Web Services (AWS)) using Apache Spark, for which we provide example scripts, including resaving into the N5 (Zarr compatible) file format (Fig. 1g and Supplementary Notes). The results are saved as a CSV file, or they can be transferred to the region-of-interest (ROI) manager for downstream analysis in Fiji (Fig. 1e and Supplementary Notes). A mask filtering tool can classify detections on the basis of a binary mask, for example a cytoplasm or nuclear mask (Supplementary Fig. SN12.1). The saved point clouds can be overlaid onto the images using Fiji[20] or BigDataViewer[26] for interactive visual inspection of even very large datasets (Fig. 1h and Supplementary Video 1).

To validate and benchmark RS-FISH, we performed quantitative comparisons against FISH-quant[14], Big-FISH[18], AIRLOCALIZE[17], Starfish[16], and deepBlink[15] using (1) simulated smFISH images with varying noise levels to assess detection performance, (2) simulated images of spot pairs that are close to one another to assess performance on dense datasets, (3) real smFISH *Caenorhabditis elegans* embryo datasets for runtime measurements, (4) real smFISH cell datasets with varying noise levels, and (5) large lightsheet datasets[13]. We show that RS-FISH is on par with the best methods in terms of detection performance. Notably, it provides high detection accuracy and low localization error (Fig. 2a–c and Supplementary Fig. SN4.1–SN7.2) while running 3.8–7.1 times faster than established methods (Fig. 2d and Supplementary Notes). We additionally compare localization error and detection accuracy across different noise levels (Fig. 2e,f). RS-FISH shows superior detection accuracy, especially in the presence of very high noise. The localization error is very good in low-noise scenarios and slightly increases for higher noise levels, which is partially explained by having to localize more spots that other methods do not detect. We provide example images of each noise class tested in Figure 2e,f as guidance for users to estimate the expected localization quality. We highlight that RS-FISH can easily be parallelized on the cloud by running smFISH extraction on 4,010 *C. elegans* image stacks (~100 GB in total) in 18 minutes on AWS at the cost of US$18.35 in June 2021 (Fig. 1g). Importantly, RS-FISH is currently the only method that can be directly applied to large volumes (Fig. 1h and Supplementary Video 1). Processing a reconstructed 148-GB lightsheet image stack took 32 CPU hours (~1 hour on a modern workstation). In comparison, a complex wrapping software for distributing AIRLOCALIZE, specifically developed for the expansion-assisted iterative FISH (EASI-FISH) project to run on the HHMI Janelia cluster, required significant development effort and took 156 CPU hours to finish the same task[13].

We developed RS-FISH based on a generic derivation of 3D RS for anisotropic objects that is efficiently implemented using ImgLib2, Fiji, and Spark. RS-FISH runs as a Fiji plugin, allowing interactive parameter adjustment and result verification on small and large images, making the task of correctly detecting diffraction-limited spots in microscopy images as accessible as possible. Processing speed is significantly improved and similar localization performance to established methods is achieved. RS-FISH is simple to install and run through Fiji, additionally providing macro-recording functionality to automate FISH spot detection easily. Our efficient block-based implementation allows easy single-molecule spot detection in large datasets or big volumes using local processing, clusters, or the cloud. Importantly, although we have demonstrated RS-FISH's utility using only a 148-GB dataset, there is no conceptual limit that prohibits RS-FISH from being executed on significantly larger volumes well into the petabyte range. RS-FISH is an accurate, easy-to-use, versatile, and scalable tool that makes FISH spot detection on small and especially large datasets amenable to

**Fig. 2 | Performance of different spot localization tools. a**, Detection accuracy for different tools was analyzed using the $F_1$ score calculated from true-positive (red circle, white spot), false-positive (red circle, no spot), and false-negative (no circle, white spot) detection. Corresponding false-positive and false-negative values can be found in Supplementary Figure SN4.2, in which data are represented as a boxplot with the full outlier range. $F_1$ score and localization error were determined using a set of 50 simulated images (256 × 256 × 32 pixels), with different noise levels (example images in Supplementary Fig. 4.1) containing either 30 spots ($n = 39$) or 300 spots ($n = 11$). The best detection parameters for each tool were determined by a grid search over the parameter space (details in Supplementary Notes). **b**, Localization error was measured as Euclidean distance (pixels) between the detected spot center and the ground-truth center of simulated spots for the same set of images described in **a**. **c**, Histograms of distance deltas of the ground truth to its corresponding localized spot separated by image dimensions ($x, y, z$) for the different tools, showing that all methods are highly accurate while precision varies. The corresponding localization error for each dimension separately can be found in Supplementary Fig. SN4.2d,e. **d**, Comparison of processing speed for 13 real 3D smFISH images of *C. elegans* embryos, with images sized around 30 MB containing an average of ~350 spots per image (example images in Supplementary Fig. SN4.1). Bar plots in **a**, **b**, and **d**, as well as the line plots in **e** and **f**, show the mean and a 95% confidence interval of the 50 measured detections. **e**, Influence of different image noise levels on spot detection. Plot displays detection accuracy measured as $F_1$ score ($y$ axis) against the s.d. of image noise ($x$ axis). Example images corresponding to the different noise levels are displayed below the graph. **f**, Influence of image noise on the localization error measured in Euclidean distance to the center of simulated points (ground truth) against the s.d. of image noise, using the same data as shown in **b**. For **a–f**, details on run parameters and tables with raw values are in Supplementary Notes.

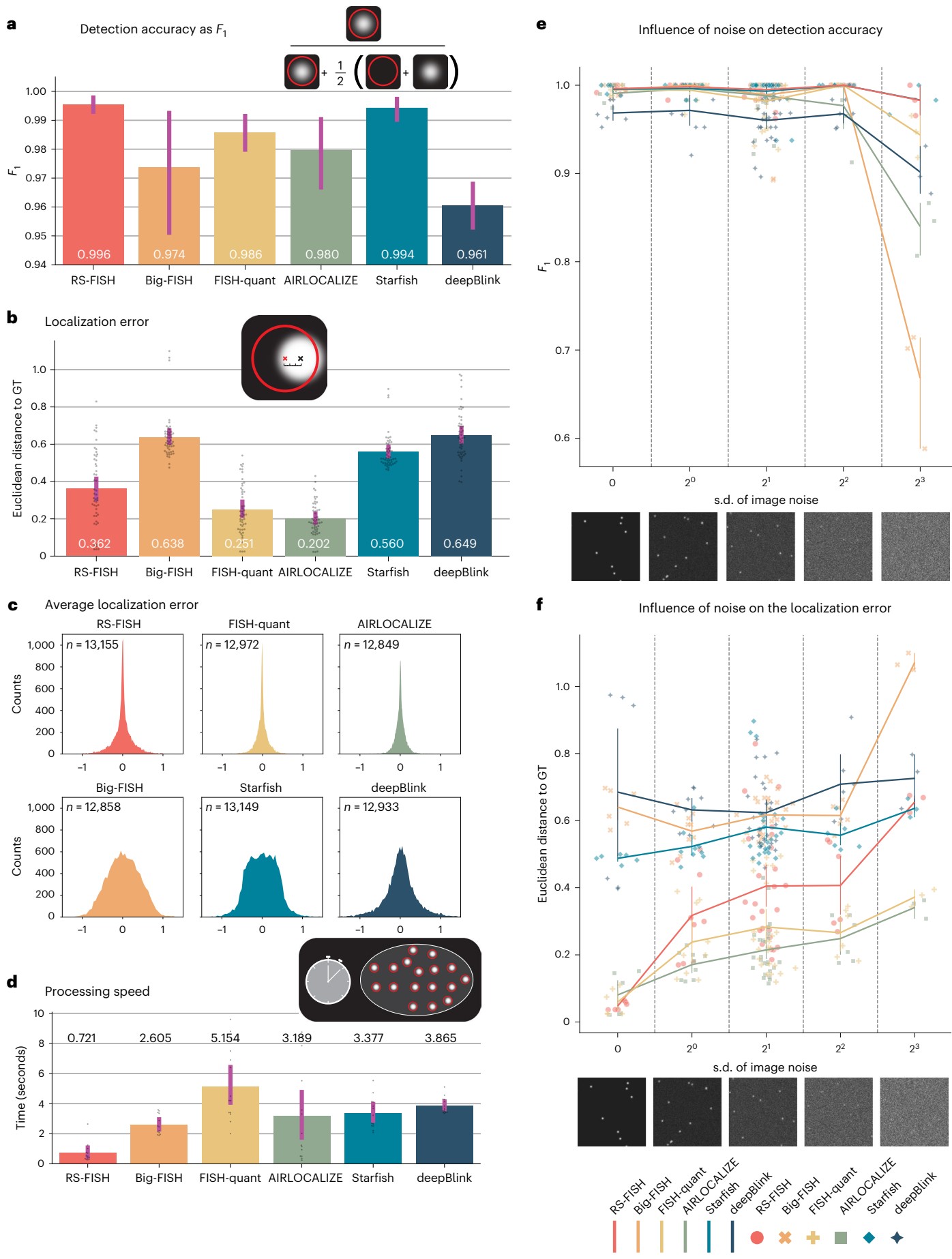

researchers and whose functionality extends to the dynamically growing fields of spatial transcriptomics and spatial genomics.

## Online content

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

## Methods

### *n*-dimensional derivation of Radial Symmetry localization

The goal of RS is to accurately localize a bright, circular spot $p_c$ with sub-pixel accuracy. In noise-free data, image gradients $\nabla I(p_k)$ at locations $p_k$ point towards the center of the spot and intersect in that single point $p_c$ (Fig. 1a), thus computing the intersection point solves the problem of accurate localization. In realistic images that contain noise, these gradients do not intersect, therefore computing $p_c$ constitutes an optimization problem that RS solved using least-squares minimization of the distances $d_k$ between the common intersection point $p_c$ and all gradients $\nabla I(p_k)$ (Supplementary Fig SN1.1).

We extend RS to 3D similar to Liu et al.[21], and additionally describe how to generalize the derivation to the *n*-dimensional case. To achieve this, we replace the Roberts cross operator with separable convolution for image gradient $\nabla I(p_k)$ computation, and we use vector algebra to compute the intersection point $p_c$ of image gradients. The derivations are shown in detail in Supplementary Fig SN1.1 and Supplementary Notes.

### Radial Symmetry for axis-aligned ellipsoid (non-radial) objects

Diffraction-limited spots in 3D microscopy images are usually not spherical but show a scaling in the axial (*z*) dimension compared with the lateral (*xy*) dimensions. Previous solutions suggested scaling the image in order to be able to detect spots using RS[21]. This can be impractical for large datasets, and it might affect localization quality, as the image intensities need to be interpolated for scaling. Here, we extend the RS derivation to directly compute the intersection point $p_c$ from anisotropic images by applying a scale vector *s* to point locations $p_k$ and applying the inverse scale vector $s^{-1}$ to the image gradients $\nabla I(p_k)$. Although we derive the case specifically for 3D, it can be straightforwardly applied to higher dimensions. The derivation is shown in detail in the Supplementary Notes.

RS-FISH supports a global scale factor (called anisotropy factor) for the entire dataset that compensates for anisotropy of the axial (*z*) dimension, which can be computed from an image containing diffraction-limited spots (Supplementary Notes).

### Radial Symmetry Random Sample Consensus

RS localization is implemented as a fast, closed-form solution, and it is therefore feasible to combine it with robust outlier removal. We use RANSAC[22] to identify the maximal number of gradients $\nabla I(p_k)$ that support the same center point $p_c$ given a maximal distance error $\varepsilon$, so that all $d_k < \varepsilon$.

To achieve this, RANSAC randomly chooses the minimal number of gradients (that is, two gradients) from the set of all gradients (candidate gradients) to compute the center point and tests how many other gradients fall within the defined error threshold $\varepsilon$. This process is repeated until the maximal set of gradients is identified (inlier gradients) and the final center point $p_c$ is computed using all inlier gradients. This allows RS-FISH to exclude artifact pixels and to differentiate close-by spots.

The number of gradients that are computed for each spot is defined by the support region radius, which can be selected as one of the RANSAC parameters. By default, we propose a radius of 3 pixels, which corresponds to a 7 × 7 × 7 pixel patch, resulting in 216 gradients for the 3D case. These settings are reasonable choices for acquisition parameters typically used for smFISH images (500–700 nm emission, ×63 oil detection objective, EMCCD or sCMOS camera with ~10-µm pixels, corresponding to a ~159-nm lateral pixel size in the sample plane), where the pixel patch comfortably covers the central peak of the point spread function (PSF). Importantly, the radius should be adjusted to the respective acquisition settings so that an area that is approximately twice the size of the central peak of the PSF is entirely covered to ensure that all gradients that point towards the center of each spot are included in the localization.

To identify and locate close-by points, RS-FISH runs a multi-consensus RANSAC. Here, RANSAC is run multiple times on the same set of candidate gradients. After each successful run that identifies a set of inliers, the inliers are removed from the set of candidate gradients, and RANSAC tries to identify another set of inliers (Fig. 1b). This process is iterated until no other set of inliers (corresponding to a FISH spot) can be found in the local neighborhood of each DoG spot. To not detect random noise, the minimal number of inliers required for a spot can be adjusted (typically around 30).

### Implementation details and limits

RS-FISH is implemented in Java using ImgLib2, the mpicbg framework, BigDataViewer, Fiji, and Apache Spark. The computation of RS is performed in blocks with a size of $b_d$ for each dimension $d$ (for example, 256 × 256 × 128 pixels) and requires an overlap of only 1 pixel in each dimension with neighboring blocks, thus the overhead $o = 1 - \frac{\prod_d b_d}{\prod_d b_d - 2}$ is minimal (for example, 1.5% for 256 × 256 × 256 blocks or 0.6% for 1024 × 1024 × 1024 blocks). When processing each block, the local process has access to the entire input image, which is either held in memory when running within Fiji or is lazy-loaded from blocked N5 datasets when running on large volumes using Apache Spark. Because the computation across blocks is embarrassingly parallel, computation time linearly scales with the dataset size. Thus, RS-FISH will run on very large volumes supported by N5 and ImgLib2. Owing to current limitations in Java arrays, the theoretical upper limit is $2^{31} = 2,147,483,648$ blocks, with each block maximally containing $2^{31} = 2,147,483,648$ pixels (for example 2048 × 2048 × 512 pixels). Given sufficient storage and compute resources, the limit for RS-FISH is thus 4,072 peta-pixels (4,072 petabytes at 8 bit, or 8,144 petabytes at 16 bit) taking into account the overhead, whereas every individual block locally processes only 2 gigapixels ($2^{31} = 2,147,483,648$ pixels).

The code can be executed on an entire image as a single block for smaller images, or in many blocks multi-threaded or distributed using Apache Spark. It is important to note that RS-RANSAC uses random numbers to determine the final localization of each spot. We use fixed seeds to initialize each block; therefore, the results for a single block of the same size in the same image with the same parameters are constant. However, for blocks of different sizes (for example, single-threaded versus multi-threaded), the results will be slightly different, as the RANSAC-based localizations are not traversing the DoG maxima in the same order, and thus initialize RANSAC differently. For practicality, the interactive Fiji mode runs only in single-threaded mode (although the DoG image is computed multi-threaded) to yield comparable results across different testing trials. Importantly, this applies only if the RANSAC mode is used for localization. Multi-threaded processing is available in the recordable advanced mode in the Fiji plugin, while the Apache Spark based distribution can be called from the corresponding RS-FISH-Spark repository.

### Data simulation for assessing localization performance

To create ground-truth datasets for assessing localization performance, we generated images simulating diffraction-limited spots in the following way: ($x,y,z$) spot positions were randomly assigned within the *z*-stack chosen dimensions, and each spot was assigned a brightness picked from a normal distribution. We computed the intensity $I(x,y,z)$ generated by each spot as follows: we first computed the predicted average number of photons received by each pixel $I_{pred}(x,y,z)$ computed using a gaussian distribution centered on the spot, with user-defined lateral and axial extensions. We then simulated the actual intensity collected at each pixel using a Poisson-distributed value with mean $I_{pred}(x,y,z)$. We eventually added gaussian-distributed noise to each pixel of the image.

Code for generating the images with simulated diffraction-limited spots is available in the GitHub repository. There is also a folder included with the simulated data used in the parameter grid search

and benchmarking: https://github.com/PreibischLab/RS-FISH/tree/master/documents/Simulation_of_data.

Additionally, we simulated a dataset that contains spots that are very close to each other in order to assess the ability of RS-FISH and other tools to differentiate such spots. The code is here: https://github.com/timotheelionnet/simulated_spots_rsFISH. The images are here: https://github.com/timotheelionnet/simulated_spots_rsFISH/tree/main/out.

### Benchmarking RS-FISH against commonly used spot-detection tools

RS-FISH performance was benchmarked against the leading tools for single-molecule spot detection in images. The tools compared in the benchmarking are FISH-quant[14] (Matlab), Big-FISH[18] (Python), AIRLO-CALIZE[17] (Matlab), Starfish[16] (Python), and deepBlink[15] (Python, Tensor-Flow). Localization performance comparison was done on simulated images with known ground-truth spot locations, and computation-time comparison was performed using real three-dimensional *C. elegans* smFISH images. We created a dedicated analysis pipeline for each tool to test localization performance and compute time. For localization performance comparison, a grid search over each tool's pipeline parameter space was run (excluding deepBlink, as a pre-trained artificial neural network was used; more details regarding deepBlink are discussed in the Supplementary Notes). Importantly, tools use different offsets for their pixel coordinates, which depend on the respective pixel origin convention (for example, does a spot positioned at the center of a pixel lie at 0.0 or 0.5? Does the *z* index start with 0 or 1?). In our benchmarks, for each tool we detected these offsets by computing the precision (the average, signed per-dimension difference between predicted and ground-truth spots) and correct for these offsets if necessary (Supplementary Fig. SN4.2d,e). RS-FISH assumes that each pixel in an image is a measurement (not a square) that is located at floored coordinates (for example 11.0, 134.0, 12.0), and the top left pixel of the first slice corresponds to the coordinates (0.0, 0.0, 0.0). For computation-time comparison on real data, each pipeline's parameters were selected to produce a similar number of detected spots for each image. Additionally, we performed benchmarks for spots that were close to each other (Supplementary Fig. SN7.1 and SN7.2 and Supplementary Notes) and on real data with varying levels of noise (Supplementary Fig. SN6.1 and Supplementary Notes).

The comparison shows that RS-FISH is on par with currently available spot-detection tools in localization performance, providing high detection accuracy and low localization error (Fig. 2a–c,e,f and Supplementary Fig. SN4.2) while running 3.8–7.1 times faster (Fig. 2d and Supplementary Notes). Additionally, RS-FISH is currently the only available tool that can be directly applied to large images, which we highlight using a 148-GB lightsheet image stack[13] (Fig. 1h, Supplementary Video 1, and Supplementary Notes). The image size of the lightsheet stack is 7,190 × 7,144 × 1,550 pixels, and the block size used for detection was 256 × 256 × 128 pixels. The detection of spots using RS-FISH took 3,263 seconds (~32 CPU hours) for the entire image on a 36 CPU workstation with 2× Intel Xeon Gold 5220 Processor at 2.2 Ghz. The runtime cannot be directly compared with the custom extension of AIRLOCALIZE that was developed for the same project, as it is written to specifically run only on the Janelia cluster. The compute time of 156 CPU hours was extracted from the cluster logs of the submission scripts and was executed on a mix of Intel SkyLake (Platinum 8168) at 2.7 GHz and Intel Cascade Lake (Gold 6248 R) at 3.0 Ghz CPUs. The overall speed increase of ~5× generally agrees with our measurements in Figure 2d, and the performance of a mix of these CPUs is comparable to the workstation CPUs (according to https://www.cpubenchmark.net). Importantly, RS-FISH runs on such volumes natively and can easily be executed on a cluster or in the cloud, thus it easily scales to significantly larger datasets. At the same time, the AIRLOCALIZE implementation is limited to the Janelia cluster, but could be extended to other LSF clusters that support job submission.

Benchmarking analysis details are in the Supplementary Notes, and all scripts and complete documentation are in the RS-FISH GitHub repository.

### Further properties of RS-FISH

Independent of the software used, localization performance is influenced by the lateral and axial sampling rate of the microscope, which has been studied extensively, for example in Thompson et al.[27]. RS-FISH supports a wide range of parameters that are explained in detail (Supplementary Fig. SN8.1) and allows the user to adjust it to the microscope settings used.

RS-FISH supports all image data formats supported by Fiji and BioFormats, including N5/Zarr. For distributed processing using Spark, large images need to be stored in the N5/Zarr format.

### Limitations of RS-FISH

RS-FISH is a tool for detecting diffraction-limited spots in single-channel 2D or 3D microscopy images. It gives the user a lot of flexibility through interactive parameter selection to detect all spots in their images. Thus, setting these parameters could potentially be daunting for new users. However, we choose default parameters that give good results for many typical FISH spot images, and the interactive GUI allows users to test out different parameters easily on their images. Other tools have limited parameter selection, but RS-FISH is able to detect spots more accurately because it allows careful, interactive parameter selection. RS-FISH precisely localizes spots in images with little noise but is less precise in images that show high noise (compared with only FISH-quant and AIRLOCALIZE), which can be partly explained by the ability of RS-FISH to correctly detect more spots in high-noise cases. Very dense spots or clouds of spots, which might be due to smFISH spots of highly expressed genes, are particularly challenging to detect using any currently available method. The multi-consensus RANSAC improves the situation, but parameter selection is not easy and it does not correct for the fact that the gradients of two very close spots influence each other (it simply ensures that the error is not higher than the user-defined RANSAC error threshold).

The RS-FISH-detected spots can be classified on the basis of their position relative to image landmarks within the plugin using binary masks. Further downstream analysis, such as co-localization, can be easily performed on the results files within Fiji or other analytical frameworks in R or Python.

### Reporting summary

Further information on research design is available in the Nature Portfolio Reporting Summary linked to this article.

## Data availability

All datasets used for benchmarking are available in the RS-FISH and Timothee Lionnet GitHub repositories (links are available in Supplementary Table 3), which includes simulations and 3D smFISH images of *C. elegans* embryos. The raw data underlying Fig. 1 are available at figshare (links are available in Supplementary Table 3).

## Code availability

RS-FISH is implemented as open-source in Java/ImgLib2 and provided as a macro-scriptable Fiji plugin and stand-alone command-line application capable of cluster and cloud execution. The code source, tutorial, documentation, and example images are available at: https://github.com/PreibischLab/RS-FISH and https://github.com/PreibischLab/RS-FISH-Spark.

## Acknowledgements

We would like to thank S. Saalfeld for providing the mpicbg framework, A. Woehler at the System Biology imaging platform for technical support, D. Friedrich for discussions about the *C. elegans*

smFISH imaging protocol, F. Mueller and A. Imbert for their advice on running Big-FISH, and J. Chao and B. Eichenberger on their referral to deepBlink's 3D solution. We are grateful to R. H. Singer (Albert Einstein College of Medicine & HHMI Janelia Research Campus) for discussions at the early stages of the project. Additional datasets for images in Figure 1 were kindly provided by H. Q. Nguyen, S. Chattoraj, T. Wu, and Y. Wang from the Multifish team project. The image of the mouse primary cortical neurons (Supplementary Fig. SN12.1) was kindly provided by S. Mendonsa and M. Chekulaeva. We would like to thank R. Klose; the mESC smFISH in Fig. 1 were labeled and imaged by L.B. in his lab. E. B. was supported by the HFSP grant RGP0021/2018-102 and MDC-Berlin, L. B. was supported by MDC-Berlin and the Joachim Herz Foundation (no. 850022), M. Z. was supported by EU H2020 Training grant 721890 'circRtrain' and HHMI Janelia, L. E. and K. I. S. H. were supported by MDC-Berlin and the Helmholtz Imaging Platform, K. K. was supported by MDC-Berlin, X. L. was supported by HHMI Janelia, T. L. and N. M. were supported by NIH grant R01 GM127538, T. L. was supported by Melanoma Research Foundation Team Science award no. 687306, B. K. was supported by NIH fellowship F32CA239394, and S. P. was supported by HFSP grant RGP0021/2018-102, EU H2020 Training grant 721890 'circRtrain', MDC-Berlin and HHMI Janelia.

## Author contributions

The project was conceived by T. L. and S. P. The mathematical concepts and derivations were done by T. L. and S. P. The software was written by S. P., E. B., and M. Z. with contributions by K. K. and K. I. S. H. The documentation was written by L. B. and K. K. The simulated data were created by T. L. The benchmarking was performed by E. B., L. B., and L. E. Experiments and imaging were performed by L. B., N. M., B.K., and X. L., K. I. S. H., T. L., and S. P. supervised the project. L. B. and S. P. wrote the manuscript with input from all co-authors.

## Competing interests

The authors declare no competing interests.

## Additional information

**Correspondence and requests for materials** should be addressed to Kyle I. S. Harrington, Timothée Lionnet or Stephan Preibisch.

# Reporting Summary

Nature Research wishes to improve the reproducibility of the work that we publish. This form provides structure for consistency and transparency in reporting. For further information on Nature Research policies, see our Editorial Policies and the Editorial Policy Checklist.

## Statistics

For all statistical analyses, confirm that the following items are present in the figure legend, table legend, main text, or Methods section.

| n/a | Confirmed | |
|---|---|---|
| ☐ | ☒ | The exact sample size (*n*) for each experimental group/condition, given as a discrete number and unit of measurement |
| ☐ | ☒ | A statement on whether measurements were taken from distinct samples or whether the same sample was measured repeatedly |
| ☒ | ☐ | The statistical test(s) used AND whether they are one- or two-sided<br>*Only common tests should be described solely by name; describe more complex techniques in the Methods section.* |
| ☒ | ☐ | A description of all covariates tested |
| ☒ | ☐ | A description of any assumptions or corrections, such as tests of normality and adjustment for multiple comparisons |
| ☐ | ☒ | A full description of the statistical parameters including central tendency (e.g. means) or other basic estimates (e.g. regression coefficient) AND variation (e.g. standard deviation) or associated estimates of uncertainty (e.g. confidence intervals) |
| ☒ | ☐ | For null hypothesis testing, the test statistic (e.g. *F*, *t*, *r*) with confidence intervals, effect sizes, degrees of freedom and *P* value noted<br>*Give P values as exact values whenever suitable.* |
| ☒ | ☐ | For Bayesian analysis, information on the choice of priors and Markov chain Monte Carlo settings |
| ☒ | ☐ | For hierarchical and complex designs, identification of the appropriate level for tests and full reporting of outcomes |
| ☒ | ☐ | Estimates of effect sizes (e.g. Cohen's *d*, Pearson's *r*), indicating how they were calculated |

*Our web collection on statistics for biologists contains articles on many of the points above.*

## Software and code

Policy information about availability of computer code

| Data collection | Micro-Manager (2.0.0), NIS Elements software,  Applied Precision SoftWoRx |
|---|---|
| Data analysis | RS-FISH (0f6ab4a), FISH-quant (v3), Big-FISH (0.5.0), AIRLOCALIZE (1.6), Starfish (0.2.2), deepBlink (0.1.1), Fiji  (2.3.0) |

For manuscripts utilizing custom algorithms or software that are central to the research but not yet described in published literature, software must be made available to editors and reviewers. We strongly encourage code deposition in a community repository (e.g. GitHub). See the Nature Research guidelines for submitting code & software for further information.

## Data

Policy information about availability of data

All manuscripts must include a data availability statement. This statement should provide the following information, where applicable:
- Accession codes, unique identifiers, or web links for publicly available datasets
- A list of figures that have associated raw data
- A description of any restrictions on data availability

All datasets used for benchmarking are available in the RS-FISH GitHub repository (https://github.com/PreibischLab/RS-FISH and https://github.com/timotheelionnet/simulated_spots_rsFISH ), which includes simulations and 3D smFISH images of C. elegans embryos. The raw data underlying Fig. 1 are available at figshare: https://doi.org/10.6084/m9.figshare.21067342.v1; https://doi.org/10.6084/m9.figshare.21067366;  https://doi.org/10.6084/m9.figshare.21067360.v1; https://doi.org/10.6084/m9.figshare.21067354; https://doi.org/10.6084/m9.figshare.21067369.v1; https://doi.org/10.6084/m9.figshare.21067372

# Field-specific reporting

Please select the one below that is the best fit for your research. If you are not sure, read the appropriate sections before making your selection.

☒ Life sciences ☐ Behavioural & social sciences ☐ Ecological, evolutionary & environmental sciences

For a reference copy of the document with all sections, see nature.com/documents/nr-reporting-summary-flat.pdf

# Life sciences study design

All studies must disclose on these points even when the disclosure is negative.

| | |
|---|---|
| Sample size | No measure were taken to estimate the sample sizes. However the sample size was limited by the computational capacity since we performed grid searches over the parameter space for each tool and analysis. The sample size was adequate as it illustrates the detection capabilities of RS-FISH compared to similar tools. The sample size of 50 images for localization performance were chosen to cover a reasonable range of simulated SNRs and point densities (each containing 30-300 points) - all can be inspected in the github repository. For the analysis of close points we created 720 images (each containing 30 points). For the benchmarks using real data we created 63 two-dimensional images with different noise levels. For speed measurements we chose 13 different samples of actual 3D image data stacks. We limited it to 13 since each tool needs to be manually tuned to yield similar numbers of points, which constitutes a major effort. |
| Data exclusions | Data in Supplementary Note 6 was excluded due to too low SNR (explained in the text). Otherwise no data was excluded. |
| Replication | Unless stated otherwise, experiments were performed once. The manuscript asses the performance of an image analysis software (mostly on simulated data) and not on variability of biological samples. |
| Randomization | The location of the points for the benchmarking datasets were chosen randomly. Data analysis was not randomized since this analysis is not affected by human bias. |
| Blinding | The benchmarking was performed and scored by computer algorithms after a grid search to find the best parameters for each tool. Since this analysis is not affected by human bias, no blinding was necessary. |

# Reporting for specific materials, systems and methods

We require information from authors about some types of materials, experimental systems and methods used in many studies. Here, indicate whether each material, system or method listed is relevant to your study. If you are not sure if a list item applies to your research, read the appropriate section before selecting a response.

### Materials & experimental systems

| n/a | Involved in the study |
|---|---|
| ☒ | ☐ Antibodies |
| ☐ | ☒ Eukaryotic cell lines |
| ☒ | ☐ Palaeontology and archaeology |
| ☐ | ☒ Animals and other organisms |
| ☒ | ☐ Human research participants |
| ☒ | ☐ Clinical data |
| ☒ | ☐ Dual use research of concern |

### Methods

| n/a | Involved in the study |
|---|---|
| ☒ | ☐ ChIP-seq |
| ☒ | ☐ Flow cytometry |
| ☒ | ☐ MRI-based neuroimaging |

# Eukaryotic cell lines

Policy information about cell lines

| | |
|---|---|
| Cell line source(s) | Cell lines were obtained from JCRB Cell Bank (JCRB0098 KURAMOCHI) |
| Authentication | As cell lines were purchased directly from JCRB, visual inspection was used to confirm morphology was consistent with description and pictures provided by the vendor. |
| Mycoplasma contamination | Cells lines tested negative for mycoplasma contamination |
| Commonly misidentified lines (See ICLAC register) | No commonly misidentified cell lines were used in the study |

## Animals and other organisms

Policy information about studies involving animals; ARRIVE guidelines recommended for reporting animal research

| | |
|---|---|
| Laboratory animals | C. elegans: N2 wild-type worms, hermaphrodite, stages mixed embryo and L2 larvae; Drosophila: strain: wild-type (w1118) ; sex: female ; stage: ZT2 |
| Wild animals | No wild animals were used in this study. |
| Field-collected samples | No field-collected samples were used in this study. |
| Ethics oversight | No ethical approval was required for Drosophila and C. elegans |

Note that full information on the approval of the study protocol must also be provided in the manuscript.

