## [Peer Review File · Nature Methods]

Peer Review Information

Manuscript Title: RS-FISH: Precise, interactive, fast, and scalable FISH spot detection

Corresponding author name(s): Kyle I. S. Harrington, Timothée Lionnet, Stephan Preibisch

Editorial Notes:

Reviewer Comments & Decisions:

Decision Letter, initial version:
--

Dear Stephan,

Your Brief Communication, "RS-FISH: Precise, interactive, fast, and scalable FISH spot detection", has now been seen by three reviewers. As you will see from their comments below, although the reviewers find your work of considerable potential interest, they have raised some concerns. We are interested in the possibility of publishing your paper in Nature Methods, but would like to consider your response to these concerns before we reach a final decision on publication.

We therefore invite you to revise your manuscript to address these concerns. We think the referee feedback is fairly clear, they would like to see more benchmarking, especially on 'real world' noisy data, and see how the method performs with dense emitters, along with other technical concerns.

There are two minor points raised by reviewer 3 that we do not think are essential. These are in regard to detecting colocalization and image registration. If these are not already features of the tool, we think these points can simply be discussed.

[Redacted] This URL links to your confidential home page and associated information about manuscripts you may have submitted, or that you are reviewing for us. If you wish to forward this email to co-authors, please delete the link to your homepage.

We hope to receive your revised paper within eight weeks (but please let me know if you need more time because of the holidays). If you cannot send it within this time, please let us know. In this event, we will still be happy to reconsider your paper at a later date so long as nothing similar has been accepted for publication at Nature Methods or published elsewhere.

OPEN SCIENCE REQUIREMENTS

REPORTING SUMMARY AND EDITORIAL POLICY CHECKLISTS

Please note that these forms are dynamic ‘smart pdfs’ and must therefore be downloaded and completed in Adobe Reader. We will then flatten them for ease of use by the reviewers. If you would like to reference the guidance text as you complete the template, please access these flattened versions at <http://www.nature.com/authors/policies/availability.html>.

DATA AVAILABILITY

We strongly encourage you to deposit all new data associated with the paper in a persistent repository where they can be freely and enduringly accessed. We recommend submitting the data to discipline-specific and community-recognized repositories; a list of repositories is provided here:

<http://www.nature.com/sdata/policies/repositories>

All novel DNA and RNA sequencing data, protein sequences, genetic polymorphisms, linked genotype and phenotype data, gene expression data, macromolecular structures, and proteomics data must be deposited in a publicly accessible database, and accession codes and associated hyperlinks must be provided in the “Data Availability” section.

Please include a “Data availability” subsection in the Online Methods. This section should inform readers about the availability of the data used to support the conclusions of your study, including accession codes to public repositories, references to source data that may be published alongside the paper, unique identifiers such as URLs to data repository entries, or data set DOIs, and any other statement about data availability. At a minimum, you should include the following statement: “The data that support the findings of this study are available from the corresponding author upon request”, describing

which data is available upon request and mentioning any restrictions on availability. If DOIs are provided, please include these in the Reference list (authors, title, publisher (repository name), identifier, year). For more guidance on how to write this section please see:
<http://www.nature.com/authors/policies/data/data-availability-statements-data-citations.pdf>

CODE AVAILABILITY

Please include a “Code Availability” subsection in the Online Methods which details how your custom code is made available. Only in rare cases (where code is not central to the main conclusions of the paper) is the statement “available upon request” allowed (and reasons should be specified).

ORCID

Nature Methods is committed to improving transparency in authorship. As part of our efforts in this direction, we are now requesting that all authors identified as ‘corresponding author’ on published papers create and link their Open Researcher and Contributor Identifier (ORCID) with their account on the Manuscript Tracking System (MTS), prior to acceptance. This applies to primary research papers only. ORCID helps the scientific community achieve unambiguous attribution of all scholarly contributions. You can create and link your ORCID from the home page of the MTS by clicking on ‘Modify my Springer Nature account’. For more information please visit www.springernature.com/orcid.

Sincerely,
Rita

Rita Strack, Ph.D.

Senior Editor
Nature Methods

Reviewers' Comments:

Reviewer #1:

Remarks to the Author:

“RS-FISH: Precise, interactive, fast, and scalable FISH spot detection” by Bahry, Breimann, & Zouinkhi et al. describes a new computational toolkit for the identification of FISH puncta in microscopy images. The core of the tool is anchored in the Fiji/ImageJ ecosystem, and extensions are provided for distributed computing on cluster and cloud architectures.

I found the manuscript clearly written, easy to follow, and well-structured. The analysis of simulated smFISH data shows clear advantages of RS-FISH compared to prior art and demonstrates its effectiveness on a panoply of experimental data from different methods/systems. The code itself is clean, easy to access, and well documented. The authors also provided detailed guidance on how to install and use their tool.

Overall, I am a big fan of this work. Although there are several tools that already exist for the segmentation of FISH spots, this is by no means a fully solved problem, particularly at scale. The ability to use RS-FISH interactively in Fiji to establish a segmentation workflow that can then be ported with minimal effort to cluster or cloud architectures is a huge plus, especially given how widely used and accessible Fiji/ImageJ is. I expect this ability will enable RS-FISH to be adopted by labs with a broad spectrum of computational abilities, which will make it of high value and interest to the large community that performs FISH experiments.

I was also impressed with how simple and effective RS-FISH was to install and use. I was able to install and run RS-FISH on a couple of DNA and RNA FISH images I had handy in under 5 minutes, and the performance was terrific—I changed one setting but otherwise kept default parameters, and the segmentation of both the DNA and RNA signals was essentially instant and flawless. Indeed, qualitatively at least, it appeared to outperform a bespoke Python pipeline we had used on the same images in our previous work. I run a quantitative imaging/FISH lab, and we will definitely be trying to integrate RS-FISH into some of our workflows based on these initial impressions of its performance.

I am thus overall very enthusiastic about this work and its fit at Nature Methods. I do have a few suggestions to improve the manuscript, which are listed below:

Major points:

1. It would be helpful for potential users to see the results of Figure 2a–c at a few different signal-to-noise levels, as the performance of the different algorithms may vary quite a bit as SNR changes. A discussion of how RS-FISH performs in the context of these results would also help provide guidance for potential users as to whether their datasets are likely to be successfully processed by RS-FISH, particular for poor SNR images.

Minor points:

1. I imagine the performance of RS-FISH is somewhat sensitive to the lateral sampling rate of the point spread function for 2D fitting and both the lateral and axial sampling rate for 3D fitting. It would be helpful for potential users to provide some guidance in the form of a figure and/or discussion about this point.

2. I think Figure 2 could be improved by including more detail in the axes and/or figure legend. I had to dig through the Supplement to fully grasp what was being presented.

Reviewer #2:

Remarks to the Author:

I've attached my comments for the authors below.

Review for manuscript NMETH-BC47378 RS-FISH: Precise, interact, fast, and scalable FISH spot detection
Recommendation: Accept with minor revisions

Key results: This paper is about a computational software called Radial Symmetry-FISH (RS-FISH). It can detect single-molecule fluorescent spots in 2D and in 3D images. It claims to be: (1) accurate, (2) robust, and (3) quick. The authors have provided sufficient evidence for all three claims in the paper. Because spot detection is used in multiple FISH applications, their computational software is widely applicable to the broader community. Their software has other features such as: (1) interactive parameter tuning and (2) the capability to scale to TB sized 3D images using distributed processing. Other than a FIJI plugin, it is also a command-line tool and a cluster/cloud distributable package for large volumes. They combined radial symmetry with Random Sample Consensus (RANSAC). RANSAC helps to remove outliers based on some user-defined localization error, separate close detections, and ignore outlier pixels that disturb localization.

Reviewer comment 1: I think beyond labeling schemes that use FISH for DNA or RNA, RS-FISH could also be applied to any method that has sparse fluorescent spots in the image data, such as single particle tracking with organic dyes, quantum dots, nanoparticles, or fluorescent proteins, or even 3D single-molecule localization microscopy. Could the authors comment on whether RS-FISH is suitable for those other applications such as single particle tracking or 3D SMLM? Or would the radial symmetry method not be accurate enough for the accuracy required in 3D SMLM?

Reviewer comment 2: Could the authors add a small statement on how big is the “local pixel patches” around each seed point? I do understand this depends on which pixels/gradients are chosen after the RANSAC step. But perhaps the authors may want to provide a few words on how the number of pixels in the local pixel patches correspond with the diffraction limit and how well the pixels are sampled? Even an example sentence such as “a typical local pixel patch size is about 11 pixels for a simulated image with 600 nm emission and 120 micron camera pixels” would help readers understand what goes into the intensity calculation later on. I do understand that there may be a limit in the main text. So maybe this sentence could go into the SI?

Reviewer comment 3:

In line 105 of the main text, the authors write that there are two methods to calculate the associated intensity value of the spot. The two methods are 1) Linear interpolation at spot’s sub-pixel location and 2) Gaussian fitting to the subset of pixels that support the spot. So, how do we choose which of the two methods are used to calculate the associated intensity value in the RS-FISH software? I may be missing this, but I do not see any option in the FIJI plugin to choose between the two methods. Could the authors indicate which is the current default option for measuring total spot intensity (maybe in the SI)?

Reviewer comment 4: Could the authors comment on how RS-FISH would perform in cases where the fluorescent spots are just too close to each other? For example, if there are 2 spots that are just too close to each other such that RS-FISH is unable to differentiate them, is the output just one really bright spot in the table of results?

Reviewer comment 5: In line 171. I suggest to change the word “round” to “spherical”. Round usually refers to 2D objects such as circles or disks.

Reviewer comment 6: I strongly suggest that the authors include a simple graphical diagram of their software workflow/pipeline for the readers and future method developers to understand their workflow. It is up to the authors to choose the design for such a graphical diagram of their workflow, but the diagram could even be just a few boxes with words and arrows. This diagram could go into the SI. From what I understand right now, it is (1) DoG for spot detection, (2) RS, (3) RANSAC, (4) Optional parameter tuning/optimization, (5) Optional background correction (6) Output results with X, Y, Z, total intensities

Reviewer comment 7: Could the authors quickly comment if the middle of the corner pixel is considered (0,0) or (0.5, 0.5) in their results table? Adding this small piece of information to the supplementary information would be helpful to some users. This sentence could perhaps go into the caption of Supp. Fig. 4.

Reviewer comment 8: Could the authors write a sentence about whether the units of the x, y, z location are in pixels? Or if RS-FISH is able to look at the metadata and give the output in units of microns? This sentence could perhaps go into the caption of Supp. Fig. 4.

Reviewer comment 9: This comment is about “positioning accuracy” or what others may refer to as “localization error”. The authors show the “Positioning accuracy” in Fig 2C. I am afraid that some readers would misinterpret Fig 2C as “all these methods are often as wrong as 0.2 to 0.3 pixels”—which would be the wrong takeaway. Shouldn’t this distribution of “localization error” be centered around 0? Is it possible for the authors to not represent the Euclidean distances as a bar chart with average and standard deviation but represent each of the 6 distributions of Euclidean distances as histograms instead? Could the authors also add into the caption of Figure 2 the number of spots that are calculated for each of the software? I can see that the authors wrote that the sample size for Fig 2a-c is 50. But is that only 50 spots for each simulation and analysis? Or is it 50 rounds of simulations with a certain number of spots in each round of simulation?

Final remarks: I think RS-FISH is a great piece of software that the bioimaging and bioimage analysis community will try to use. I have tried using their software on my own image data and it works. It is fast and easy to use. I know of several research groups who are already using it for their analysis or are considering integrating it into their image analysis workflow. I will use it for fast analysis of RNA FISH data (where the further downstream analysis and biological interpretation of the data can tolerate lower localization precision), but will not use it for high-precision analysis of single-particle tracking or single-molecule localization microscopy data. One particular FISH application that I would not use RS-FISH for is DNA FISH (i.e., chromosome walking) where the 3D localization precision is really critical to understanding the nanoscale conformation of DNA in chromosomes. Many thanks to the authors for creating this piece of software, preprinting it, writing a great tutorial, and making their software available on GitHub.

Reviewer #3:

Remarks to the Author:

Bahry et al. present a new tool, RS-FISH, for detection of single molecule FISH spots in microscopy images. They demonstrate its utility on a variety of datasets across different organisms, data types, and images. They further benchmark the tool against 5 existing softwares and provide code and training resources for users. The most novel aspects of their tool are 1) the correction of anisotropy in z stacks and 2) the use of intensity gradients and radial symmetry to identify individual spots. While the tool is working well, the benchmarking could be strengthened. The advantage of this tool compared to existing approaches could also be better highlighted.

Major comments:

Benchmarking for accuracy was done only with simulated data. The authors should provide a comparison of spot detection/accuracy using real data, preferably different kinds of data with different types of noise. While it's appreciated that computation time comparison was done on real smFISH images, localization comparisons should also be shown on real data.

A standard test like 'F1 score' can be used to compare the 'False positive' and 'False negative' rates across different methods. Segmentation results from all methods on a single sample image can be shown and a metric like 'Dice Coefficient' or 'Jaccard Index' can be used to evaluate the results. For example, see <https://www.biorxiv.org/content/10.1101/2021.03.01.433040v1.full>

Minor comments:

The authors acknowledge that performance is better for dyes in the far red range due to issues with autofluorescence. How does the software perform for samples with high autofluorescence. Is there a difference in performance based on dyes--for example something in the 488 range where autofluorescence is higher?

Does the software perform well on images from human tissue? What about complex tissues such as postmortem human brain with high autofluorescence due to lipofuscin pigments.

What file formats does the software accept? This should be noted somewhere in the article.

Is the software able to perform any co-localization with other fluorophores. For example, can you count how many spots are in a DAPI-positive nucleus or look at spots within an ROI?

How does the software perform for very highly expressed genes (i.e. cell type markers) where individual spots substantially overlap due to a high density of transcripts? Can multi-consensus RANSAC handle images where the majority of spots are not distinct puncta? This is often the case for smFISH data using RNAscope V2 technology where fluorescence amplification can cause neighboring spots of highly expressed genes to be substantially overlapping and appear more like immunofluorescence staining as opposed to spots.

The software does not appear to offer multi-channel detection. This may be important to users who are generating multiplex data in a single file. Currently users would have to take a multiplex image and save each channel individually and process, which requires additional time and data storage requirements. Is multi-channel detection a feature that the authors can add to the software to improve usability?

How are overlapping dots with the same intensity gradient and size differentiated?

The article should briefly mention limitations of the software (for example, limitations for multi-channel images, co-localization, etc.).

Figure 1: a) what do the white lines in the middle image indicate. b) title of the image is misspelled 'Two around truth points'. Should this be "ground" truth?

a. Importantly, RS-FISH is currently the only method that can be directly applied to large volumes. Can the authors justify the above sentence with all other methods mentioned in the paper, specially the starFISH tool.

b. Processing a reconstructed 148 GB lightsheet image stack took 32 CPU hours (~1 hour on a modern workstation). In comparison, a complex wrapping software for distributing AIRLOCALIZE, specifically developed for the EASI-FISH project to run on the HHMI Janelia cluster, required significant development effort and took 156 CPU hours to finish the same task.

Same as above, was the comparison made with any other tool, apart from AIRLOCALIZE?

Does the tool provide any registration functions for spatiotemporal or spatial transcriptomics methods, where the dots are detected from multiple imaging rounds. Also, does the position accuracy have any effect on these datasets, as this metric (supplementary table 1. Euclidean distance) is less accurate compared to other tools.

Parameter tuning is user dependent and having a lot of parameters to tune is a limitation that should be noted. We tested multi-consensus RANSAC on some of our multi channel smFISH images, but no dots were detected. We were not sure how to optimize settings to achieve detection. The authors should provide more sample data and details in the tutorial on how to use these different options on different images. Details on installation on different OS for batch processing should also be provided.

Author Rebuttal to Initial comments

We thank the reviewers for their thoughtful assessment of our spot detection method and our manuscript. Please see our response to each comment summarizing and highlighting the changes to the text, as well as the additional benchmarking and explanations. Throughout the text below, the original reviewer comments are provided in black, and our response is in purple. In the main text and supplement of our revised manuscript, all modified portions are highlighted in red.

Reviewer #1:

Remarks to the Author:

“RS-FISH: Precise, interactive, fast, and scalable FISH spot detection” by Bahry, Breimann, & Zouinkhi et al. describes a new computational toolkit for the identification of FISH puncta in microscopy images. The core of the tool is anchored in the Fiji/ImageJ ecosystem, and extensions are provided for distributed computing on cluster and cloud architectures.

I found the manuscript clearly written, easy to follow, and well-structured. The analysis of simulated smFISH data shows clear advantages of RS-FISH compared to prior art and demonstrates its effectiveness on a panoply of experimental data from different methods/systems. The code itself is clean, easy to access, and well documented. The authors also provided detailed guidance on how to install and use their tool.

Overall, I am a big fan of this work. Although there are several tools that already exist for the segmentation of FISH spots, this is by no means a fully solved problem, particularly at scale. The ability to use RS-FISH interactively in Fiji to establish a segmentation workflow that can then be ported with minimal effort to cluster or cloud architectures is a huge plus, especially given how widely used and accessible Fiji/ImageJ is. I expect this ability will enable RS-FISH to be adopted by labs with a broad spectrum of computational abilities, which will make it of high value and interest to the large community that performs FISH experiments.

I was also impressed with how simple and effective RS-FISH was to install and use. I was able to install and run RS-FISH on a couple of DNA and RNA FISH images I had handy in under 5 minutes, and the performance was terrific—I changed one setting but otherwise kept default parameters, and the segmentation of both the DNA and RNA signals was essentially instant and flawless. Indeed, qualitatively at least, it appeared to outperform a bespoke Python pipeline we had used on the same images in our previous work. I run a quantitative imaging/FISH lab, and we will definitely be trying to integrate RS-FISH into some of our workflows based on these initial impressions of its performance.

I am thus overall very enthusiastic about this work and its fit at Nature Methods. I do have a few suggestions to improve the manuscript, which are listed below:

We thank the reviewer for their very kind words. We are excited to hear that the reviewer successfully used RS-FISH and that it could be helpful in their future experiments.

Major points:

1. It would be helpful for potential users to see the results of Figure 2a–c at a few different signal-to-noise levels, as the performance of the different algorithms may vary quite a bit as SNR changes. A discussion of how RS-FISH performs in the context of these results would also help provide guidance for potential users as to whether their datasets are likely to be successfully processed by RS-FISH, particularly for poor SNR images.

We agree that considering different SNRs can help differentiate the different tools and, therefore, we included an analysis using simulated data with sorted SNR. In Fig. 2e, we plotted the detection accuracy as F1 score for different standard deviations of image noise. Corresponding simulated images are displayed

below the plot and are shown in supplemental figure 2a. The analysis shows that RS-FISH has a high detection accuracy for all tested SNR. Most tools have significantly reduced detection accuracy, especially at higher noise levels where RS-FISH and Starfish perform best. Fig. 2f shows the localization accuracy measured as Euclidean distance to the center of the simulated ground-truth data. In images with low noise levels, RS-FISH has a small localization error, comparable with the results of FISH-quant and AIRLOCALIZE, which use Gaussian fitting methods. For very high noise levels, the localization error of RS-FISH is comparable with Starfish, which is partially explained by the fact that RS-FISH finds more spots that other methods do not detect; those spots likely lie at the lower end of the SNR range and thus bias towards larger errors (we estimated this effect at ~10%). We additionally added small pictures for each noise class studied so users are able to get an idea of what quality of localization to expect (by all benchmarked methods). Throughout all noise levels RS-FISH shows a very good detection accuracy; i.e. finds most of the ground-truth spots.

Additionally, we included an analysis of the average localization precision in 3D in Fig. 2c and separated it into X, Y, and Z in supplemental figures 2c (without noise) and 2d (with noise). This analysis shows that the detection precision of all methods is practically 0, meaning there are no systematic errors in any of the methods.

Minor points:

1. I imagine the performance of RS-FISH is somewhat sensitive to the lateral sampling rate of the point spread function for 2D fitting and both the lateral and axial sampling rate for 3D fitting. It would be helpful for potential users to provide some guidance in the form of a figure and/or discussion about this point.

Considering the sampling rate is a great point that we now mention in the Methods section under "Further properties of RS-FISH". As this affects any localization method, it is widely studied and theoretically understood - we now cite relevant literature (<https://www.sciencedirect.com/science/article/pii/S000634950275618X>) that can guide the right choice of microscopy technique, objective, and settings. Additionally, we created a detailed explanation of RS-FISH parameters (Supplemental Figure 7) that helps to adjust the software to the respective microscopy setting used to acquire the images. We also included in the "**Choosing the right parameters for RS-FISH**" section guidelines to adjust the distance-unit parameters as a function of the sigma (a proxy for the sampling rate) and noise.

2. I think Figure 2 could be improved by including more detail in the axes and/or figure legend. I had to dig through the Supplement to fully grasp what was being presented.

We thank the reviewer for this feedback. We agree and thus added more details to the figure legend in Fig. 2 to facilitate the understanding. We hope the new version now includes all the necessary details.

Reviewer #2:

Recommendation: Accept with minor revisions

Key results: This paper is about a computational software called Radial Symmetry-FISH (RS-FISH). It can detect single-molecule fluorescent spots in 2D and in 3D images. It claims to be: (1) accurate, (2) robust, and (3) quick. The authors have provided sufficient evidence for all three claims in the paper. Because spot detection is used in multiple FISH applications, their computational software is widely applicable to the broader community. Their software has other features such as: (1) interactive parameter tuning and (2) the capability to scale to TB sized 3D images using distributed processing. Other than a FIJI plugin, it is also a command-line tool and a cluster/cloud distributable package for large volumes. They combined radial symmetry with Random Sample Consensus (RANSAC). RANSAC helps to remove outliers based on some user-defined localization error, separate close detections, and ignore outlier pixels that disturb localization.

We thank the reviewer for their positive comments on our software and constructive input!

Reviewer comment 1: I think beyond labeling schemes that use FISH for DNA or RNA, RS-FISH could also be applied to any method that has sparse fluorescent spots in the image data, such as single particle tracking with organic dyes, quantum dots, nanoparticles, or fluorescent proteins, or even 3D single-molecule localization microscopy. Could the authors comment on whether RS-FISH is suitable for those other applications such as single particle tracking or 3D SMLM? Or would the radial symmetry method not be accurate enough for the accuracy required in 3D SMLM?

That is an excellent point, and we agree that it can be used for localizing any kind of diffraction-limited spot. Specifically, single-particle tracking approaches using fluorescent proteins or dyes should be able to use RS-FISH for localization as it is exactly the same problem – as long as no motion-blur effects occur that would violate the assumption of the spots being symmetric. We state in the final paragraph that RS-FISH is made for “diffraction-limited spots in microscopy images”, which is also the basis of all benchmarks shown in the paper.

3D single-molecule localization microscopy, however, is a somewhat unique case. People do use SMLM software for smFISH, and I am sure vice versa (e.g. see here: https://twitter.com/Maurice_Y_Lee/status/1504441870105014276?s=20&t=TkblleWW8dolj1LzAWdfWg). However, there are specific challenges to each problem, and different features might be relevant for each type of microscopy. For example, smFISH usually has a high background signal because many thick tissue samples have autofluorescence and/or retain low level of non-specifically bound FISH probes. In contrast, SMLM usually has less due to different imaging approaches like TIRF and HILO or the properties of the fluorophores like blinking. Additionally, for smFISH, counting the number of transcripts correctly (which RS-FISH is specifically good at) can be more critical than localizing them as accurately as possible. Additionally, precisely characterizing the PSF is crucial for as-accurate-as-possible 3D localization, which RS-FISH does not perform. Therefore, we do not want to claim that RS-FISH is a good choice for SMLM.

Reviewer comment 2: Could the authors add a small statement on how big is the “local pixel patches” around each seed point? I do understand this depends on which pixels/gradients are chosen after the RANSAC step. But perhaps the authors may want to provide a few words on how the number of pixels in the local pixel patches correspond with

the diffraction limit and how well the pixels are sampled? Even an example sentence such as “a typical local pixel patch size is about 11 pixels for a simulated image with 600 nm emission and 120 micron camera pixels” would help readers understand what goes into the intensity calculation later on. I do understand that there may be a limit in the main text. So maybe this sentence could go into the SI?

This is a great suggestion, thanks so much! We added more explanation of this central parameter to the Methods part under “Radial Symmetry RANSAC”.

Reviewer comment 3: In line 105 of the main text, the authors write that there are two methods to calculate the associated intensity value of the spot. The two methods are 1) Linear interpolation at spot’s sub-pixel location and 2) Gaussian fitting to the subset of pixels that support the spot. So, how do we choose which of the two methods are used to calculate the associated intensity value in the RS-FISH software? I may be missing this, but I do not see any option in the FIJI plugin to choose between the two methods. Could the authors indicate which is the current default option for measuring total spot intensity (maybe in the SI)?

Thanks so much for catching this. We added Gaussian intensity fit back in as an option in the main dialog. Linear interpolation is the default mode, which we also clarified in the main text.

Reviewer comment 4: Could the authors comment on how RS-FISH would perform in cases where the fluorescent spots are just too close to each other? For example, if there are 2 spots that are just too close to each other such that RS-FISH is unable to differentiate them, is the output just one really bright spot in the table of results?

We thank the reviewers for this very relevant question. To answer this, we generated a set of simulated images (n= 720) with 30 diffraction-limited points in pairs of two with different distances between the paired points (from 0.25 - 4 pixels) and different SNR as shown in Supplemental Figure 5 and 6. RS-FISH achieves the best detection accuracy compared to all tools using the implemented multi-consensus RANSAC feature. Multi-consensus RANSAC tries to detect spots using several rounds of RANSAC to identify all possible sets of inliers from a set of candidate gradients. Especially the low percent of false-negative detections illustrates that RS-FISH does a good job finding very close points. The localization error is however increased for RS-FISH since the tool detects more spots overall in the images and gradients from close points influence each other. Nevertheless, this illustrates that detection of very close points is more challenging but that RS-FISH is a good solution when the goal is to detect all spots in the image correctly. To help the users to correctly choose parameters for multi-consensus RANSAC we added “8. Choosing the right parameters for RS-FISH” to the Supplementary Note.

Reviewer comment 5: In line 171. I suggest to change the word “round” to “spherical”. Round usually refers to 2D objects such as circles or disks.

Great point! We changed “round” to “spherical”.

Reviewer comment 6: I strongly suggest that the authors include a simple graphical diagram of their software workflow/pipeline for the readers and future method developers to understand their workflow. It is up to the authors to choose the design for such a graphical diagram of their workflow, but the diagram could even be just a few boxes

with words and arrows. This diagram could go into the SI. From what I understand right now, it is (1) DoG for spot detection, (2) RS, (3) RANSAC, (4) Optional parameter tuning/optimization, (5) Optional background correction (6) Output results with X, Y, Z, total intensities

We thank the reviewer for this great suggestion! We agree that including a graphical diagram would help users choose the best workflow and parameters within RS-FISH and included a diagram in the SI as Supplemental Figure 7.

Reviewer comment 7: Could the authors quickly comment if the middle of the corner pixel is considered (0,0) or (0.5, 0.5) in their results table? Adding this small piece of information to the supplementary information would be helpful to some users. This sentence could perhaps go into the caption of Supp. Fig. 4.

We mentioned in the Methods part that we correct for differences in pixel origin conventions for the benchmarks, which was probably not clear enough. We now explicitly state that RS-FISH uses 0.0, 0.0, 0.0 as the origin; i.e. each pixel is assumed to be a measurement located at a certain position, the first one being 0.0, 0.0, 0.0. We additionally added it to Supplemental Figure 10 (previously 4).

Reviewer comment 8: Could the authors write a sentence about whether the units of the x, y, z location are in pixels? Or if RS-FISH is able to look at the metadata and give the output in units of microns? This sentence could perhaps go into the caption of Supp. Fig. 4.

Pixel locations are in raw pixel units. We chose to do that because the metadata of images saved by commercial and home-built microscopes is often not correctly populated when loaded using open-source tools. This function could be added to the tool, we just fear that it creates more harm than good. If the reviewers strongly disagree, we are happy to change it. We do mention this fact now in Suppl. Figure 10.

Reviewer comment 9: This comment is about “positioning accuracy” or what others may refer to as “localization error”. The authors show the “Positioning accuracy” in Fig 2C. I am afraid that some readers would misinterpret Fig 2C as “all these methods are often as wrong as 0.2 to 0.3 pixels”—which would be the wrong takeaway. Shouldn’t this distribution of “localization error” be centered around 0? Is it possible for the authors to not represent the Euclidean distances as a bar chart with average and standard deviation but represent each of the 6 distributions of Euclidean distances as histograms instead? Could the authors also add into the caption of Figure 2 the number of spots that are calculated for each of the software? I can see that the authors wrote that the sample size for Fig 2a-c is 50. But is that only 50 spots for each simulation and analysis? Or is it 50 rounds of simulations with a certain number of spots in each round of simulation?

This is a very valid concern indeed. We now added one plot to figure 2 showing a histogram of the distance between ground truth spot and detection for each tool, which is centered around 0. More detailed plots in supplemental figure 3 show the distribution of the error in X, Y, and Z for simulated images with and without noise. This analysis illustrates that all methods are accurate, while precision varies. We also mention this in the legend of Fig. 2.

We also added the information about the number of spots analyzed to the figure legend. We simulated 50 images in total, 39 images had 30 points and 11 images had 300 points, therefore, the total number of analyzed points is 5640.

Final remarks: I think RS-FISH is a great piece of software that the bioimaging and bioimage analysis community will try to use. I have tried using their software on my own image data and it works. It is fast and easy to use. I know of several research groups who are already using it for their analysis or are considering integrating it into their image analysis workflow. I will use it for fast analysis of RNA FISH data (where the further downstream analysis and biological interpretation of the data can tolerate lower localization precision), but will not use it for high-precision analysis of single-particle tracking or single-molecule localization microscopy data. One particular FISH application that I would not use RS-FISH for is DNA FISH (i.e., chromosome walking) where the 3D localization precision is really critical to understanding the nanoscale conformation of DNA in chromosomes. Many thanks to the authors for creating this piece of software, preprinting it, writing a great tutorial, and making their software available on GitHub.

We thank the reviewer for their kind remarks about RS-FISH and are excited to hear that our tool was already successfully used on their data. We added a limitations paragraph in the methods section to address the strengths and limitations of our tool. However, on regular DNA FISH, RS-FISH can and has been used successfully as shown in this recent paper by Frank *et al.*¹ detecting DNA FISH spots at the telomeres.

We now additionally added plots (Fig. 2f) that show the relationship between the noise level and localization error, which illustrates that localization error increases in RS-FISH only once the noise level increases. So you can comfortably use RS-FISH for any analysis if you have a good signal.

On a side note, each method only has to localize the spots it finds correctly - and RS-FISH finds more spots correctly in noisy images, which it then has to localize. So, we expect some percentage of the drop (not all of it) in localization accuracy is due to the fact that RS-FISH finds more spots - which is hard to test across all tools systematically.

Reviewer #3:

Remarks to the Author:

Bahry et al. present a new tool, RS-FISH, for detection of single molecule FISH spots in microscopy images. They demonstrate its utility on a variety of datasets across different organisms, data types, and images. They further benchmark the tool against 5 existing softwares and provide code and training resources for users. The most novel aspects of their tool are 1) the correction of anisotropy in z stacks and 2) the use of intensity gradients and radial symmetry to identify individual spots. While the tool is working well, the benchmarking could be strengthened. The advantage of this tool compared to existing approaches could also be better highlighted.

¹ Frank, L. *et al.* ALT-FISH quantifies alternative lengthening of telomeres activity by imaging of single-stranded repeats. *Nucleic Acids Res* (2022) doi:10.1093/nar/gkac113.

We thank the reviewer for their very constructive input. We think that we significantly strengthened the benchmarking aspect of the publication and highlighted its novel aspects more in the abstract. On top of the mentioned novel aspects, RS-FISH (1) is currently the only tool that provides high localization accuracy and low localization error in the same tool, (2) is significantly faster than existing solutions (3) is the only tool that provides an interactive GUI for small and large volumetric datasets and (4) is the only tool that scales to very large datasets. We highlighted these strengths throughout the abstract and text.

Major comments:

Benchmarking for accuracy was done only with simulated data. The authors should provide a comparison of spot detection/accuracy using real data, preferably different kinds of data with different types of noise. While it's appreciated that computation time comparison was done on real smFISH images, localization comparisons should also be shown on real data.

We focused on simulated data (with different noise levels) for most benchmarks because it allows using ground-truth to accurately test if the tools were over or under detecting spots and to assess the subpixel localization error to the ground-truth center. As also pointed out by reviewer 2, all tools for smFISH are effectively methods for localizing diffraction-limited spots. Using real data does not allow to precisely determine either detection accuracy (F1) or localization error due to the lack of such ground-truth information. Additionally, the physics behind the appearance of diffraction-limited spots in microscopy is well understood and can be simulated. This is supported by how algorithms are assessed for related tasks such as single-molecule localization microscopy (<https://srm.epfl.ch/Challenge> and <https://www.nature.com/articles/nmeth.3442>). We do agree that experimental data is extremely useful to convincingly show that smFISH itself works (which is luckily not a concern anymore), but it is sub-optimal for assessing point localization metrics across tools.

Nevertheless, we of course still tried to address the concern of the reviewer. To analyze how real image noise influences the detection we imaged cells labeled with smFISH probes with different acquisition settings for the same field of view (Supplemental Figure 4). Then we defined a set of points that are detected in all tools (excluding deepBlink as previous results were significantly worse) in the image with the best SNR and treated those as some sort of ground-truth points. However, it is important to realize that the actual location of this set of points is unknown and that they are biased – it is unclear to us which characteristics points share that are detected across all methods (e.g. it might be clustered spots).

Using this very approximate ground-truth information, we performed a grid-search on the parameter space for each tool to find only these spots in the images with worse SNR (keep in mind that in practice one can really only do this effectively with RS-FISH due to the interactive parameter selection). Using this approach, we are only able to report false-negative detections since we cannot assess if additionally detected points are true spots or false-positive detections, and we cannot measure the localization error. Additionally, optimizing the detections to have as few as possible false-negative detections can also skew the detection into overdetection. Nevertheless, this data shows RS-FISH can detect spots well also in real data with low SNR (Supplementary Fig. 4).

A standard test like 'F1 score' can be used to compare the 'False positive' and 'False negative' rates across different methods. Segmentation results from all methods on a single sample image can be shown and a metric like 'Dice Coefficient' or 'Jaccard Index' can be used to evaluate the results. For example, see <https://www.biorxiv.org/content/10.1101/2021.03.01.433040v1.full>

That is a great point! We now added the F1 score analysis to Fig. 2 and moved the false-positive and false-negative analysis to the supplemental material. We actually found a minor bug in our test routine when updating the scores that did not significantly impact the results but we have now corrected it. The numbers are now double-checked and nicely show two plots (Fig 2a,b) comparing localization accuracy (number of spots correctly found) and localization error (how well each method localizes the identified spots).

Minor comments:

The authors acknowledge that performance is better for dyes in the far red range due to issues with autofluorescence. How does the software perform for samples with high autofluorescence. Is there a difference in performance based on dyes--for example something in the 488 range where autofluorescence is higher?

Many biological samples have high autofluorescence in wavelengths around 488 nm and therefore dyes that overlap with this wavelength have a lower SNR as the autofluorescence adds signal to the background². Dye performance and/or autofluorescence properties are highly sensitive to the nature of the fluorophore, the origin of each individual sample (e.g. autofluorescence is often higher in aged tissue), and to the protocol used for fixation, permeabilization, and mounting prior to imaging. Since those features are different for every user, we cannot make a general statement. However, localization accuracy is generally affected by different SNR for all detection methods (see Ober *et al.*³, Kay⁴ and FISH-quant⁵ supplemental figure S4). We added Figures 2e and 2f to analyze the effect of simulated noise on the detection accuracy and the localization error. RS-FISH shows superior detection accuracy, especially in the presence of very high noise. The localization error is very good in low-noise scenarios, and slightly increases for higher noise levels, which is partially explained by having to localize more spots that other methods do not detect.

Does the software perform well on images from human tissue? What about complex tissues such as postmortem human brain with high autofluorescence due to lipofuscin pigments.

² Jennifer C. Waters; Accuracy and precision in quantitative fluorescence microscopy. *J Cell Biol* 29 June 2009; 185 (7): 1135–1148

³ Ober, R. J., Ram, S. & Ward, E. S. Localization Accuracy in Single-Molecule Microscopy. *Biophys J* 86, 1185–1200 (2004).

⁴ Kay, S. M. *Fundamentals of Statistical Signal Processing Vol. 2: Detection Theory* (Prentice Hall PTR, 1998).

⁵ Mueller, F. *et al.* FISH-quant: automatic counting of transcripts in 3D FISH images. *Nature Methods* **10**, 277–278 (2013).

Complex samples with high autofluorescence signals can lead to false-positive detections and is indeed an important problem (which is not limited to RS-FISH). There are several strategies how to overcome this problem using RS-FISH:

1. Creating a mask to only count spots within a segmented structure to exclude false detections outside cells (here for example cultured neurons). We added a mask filtering tool to RS-FISH that can segment the results table based on a binary file mask (see Supplementary Note “Segmenting detections with a binary mask”).

2. Detecting spots like lipofuscin spots or other granular/round spots separately using RS-FISH and subtracting these detections from the total FISH detections (the images shown are examples from the dotdotdot⁶ paper). This method relies on the ability to separate the signal of the autofluorescent spots (for instance lipofuscin) using spectral unmixing as nicely shown in the dotdotdot paper.

⁶ Maynard, K. R. *et al.* dotdotdot: an automated approach to quantify multiplex single molecule fluorescent in situ hybridization (smFISH) images in complex tissues. *Nucleic Acids Res* **48**, gkaa312- (2020).

- Setting the RS-FISH parameters to differentiate true signal from background via size or intensity. (See supplemental figure 6, shown example images here from the dotdotdot⁶ paper).

Lipofuscin detection

x	y	z	intensity	
121.4382	106.0834	5.0612	1	5874.5840
44.8896	113.5664	5.5437	1	4230.3483
173.4492	148.7633	2.8277	1	523.3461
172.9486	134.2773	2.3083	1	3748.3255
132.3699	168.0636	3.3982	1	2437.3075
174.2878	185.9133	3.3367	1	4999.0497
256.8163	80.3884	2.6854	1	1575.4843
234.0152	91.1649	3.2132	1	1541.8027
309.4825	54.3243	3.7639	1	873.6231
180.2192	157.0549	2.3466	1	3872.0503
253.8453	162.5443	2.6795	1	2320.4676
210.9223	184.4966	3.7373	1	942.6801

mRNA detection

x	y	z	intensity	
131.154	13.356	1.828	1	491.293
71.827	82.486	2.037	1	616.230
66.109	82.529	3.039	1	522.403
184.277	21.463	2.776	1	809.886
112.134	17.829	2.843	1	565.963
129.882	107.204	3.387	1	1749.247
131.049	164.913	2.795	1	268.437
143.030	164.343	2.389	1	419.678
172.491	130.875	3.777	1	649.237
153.825	170.953	3.089	1	1050.812
43.573	134.884	4.108	1	176.322
44.732	109.204	4.287	1	442.263
139.246	168.148	3.845	1	497.238
173.601	136.233	3.863	1	553.039
176.911	137.337	2.467	1	637.932
180.478	135.076	2.839	1	883.547
254.468	90.395	2.475	1	722.340

What file formats does the software accept? This should be noted somewhere in the article.

We added to the text that RS-FISH supports all image data formats supported by Fiji and BioFormats, including N5/Zarr. For distributed processing using Spark, large images need to be stored in the N5/Zarr format.

Is the software able to perform any co-localization with other fluorophores. For example, can you count how many spots are in a DAPI-positive nucleus or look at spots within an ROI?

While our software only detects spots within each image separately, it is relatively easy to compare results between images since our output file contains the position information for each detection. For co-localization analysis between two mRNA types, the detection lists can be compared. To filter spots that are only within a nucleus or cell, masks can be created using segmentation tools like ilastik, cellpose, or stardist and used to sort out detections outside the marked areas. We included a masking script to our plugin that filters detections based on a binary image (see Supplementary Note “Segmenting detections with a binary mask”).

How does the software perform for very highly expressed genes (i.e. cell type markers) where individual spots substantially overlap due to a high density of transcripts? Can multi-consensus RANSAC handle images where the majority of spots are not distinct puncta? This is often the case for smFISH data using RNAscope V2 technology where fluorescence amplification can cause neighboring spots of highly expressed genes to be substantially overlapping and appear more like immunofluorescence staining as opposed to spots.

This is a great point, thanks so much for bringing this up. To answer this, we generated a set of simulated images (n=720) each containing 30 diffraction-limited spots, always in pairs of two. They show different distances between the paired spots (from 0.25 - 4 pixels) and different noise levels as shown in Supplemental Figure 5 and 6.

Multi-consensus RANSAC attempts to detect spots using several rounds of RANSAC to identify all possible sets of inliers from a set of candidate gradients, thus being able to better separate close points (Fig. 1b). We again ran a grid search over all tools. RS-FISH achieves the best detection accuracy compared to all tools using the implemented multi-consensus RANSAC feature. Especially the low percent of false-negative detections illustrates that RS-FISH does a reasonable job of finding very close points.

The localization error is, however, slightly increased for RS-FISH since the tool detects more spots overall in the images and because the gradients of close spots can influence each other. This illustrates that detection of very close points is more challenging and that RS-FISH is a good solution when the goal is to detect as many spots as possible in the image correctly.

In order to guide the user to set the parameters correctly, we added instructions to the Supplemental Material (“Choosing the right parameters for RS-FISH”). Additionally, we added a discussion regarding the multi-consensus RANSAC to the limitations section.

The software does not appear to offer multi-channel detection. This may be important to users who are generating multiplex data in a single file. Currently users would have to take a multiplex image and save each channel individually and process, which requires additional time and data storage requirements. Is multi-channel detection a feature that the authors can add to the software to improve usability?

We are sorry, the current warning message is indeed confusing. It is actually much easier, one simply needs to select “Image > Color > Split Channels”, which creates as many images/image stacks as there are channels without using any additional memory. RS-FISH can then be run individually on each channel. For processing many stacks, this can easily be Macro-recorded.

We now changed the message to be clearer: “Multichannel image detected. Please select 'Image > Color > Split Channels' and run RS-FISH on each channel separately (they very likely need different parameters).” -

We chose that path because running it on the currently active channel can easily lead to wrong results in certain display modes (Composite modes), where it can be not very obvious which channel is currently selected.

How are overlapping dots with the same intensity gradient and size differentiated?

If two (or more) dots are in the exact same location, they cannot be distinguished since it is ambiguous - it could be one bright dot or many less bright ones. If there is knowledge about the expected brightness of each dot, they could be separated later by analyzing the intensities from the results table.

If two spots are very close to each other, the gradients in between the two dots are rejected by RANSAC since they do not point to either center (within the error margin defined by the *max error* parameter).

The article should briefly mention limitations of the software (for example, limitations for multi-channel images, co-localization, etc.).

A great point! Sorry, we forgot that. We agree that pointing out limitations in new tools is very helpful for users, therefore we added a section on limitations of the software to the methods.

Figure 1: a) what do the white lines in the middle image indicate. b) title of the image is misspelled ‘ Two around truth points’. Should this be “ground” truth?

a) The white lines indicate the intensity gradients that were excluded due to the RANSAC outlier removal as they were not within the defined error (green dotted circle). We included a description of the white lines in the figure caption for clarity.

b) We thank the reviewer for spotting this mistake that was caused by overlapping images resulting in a cropping of the text. We corrected this mistake.

a. Importantly, RS-FISH is currently the only method that can be directly applied to large volumes. Can the authors justify the above sentence with all other methods mentioned in the paper, specially the starFISH tool.

All other methods, including Starfish, only support in-memory processing - meaning that the input images and any temporary data structures need to fit into RAM while processing. We tried to run Starfish on the 148GB EASI-FISH sample using a 512GB RAM machine and it still failed. We were also surprised by it and reached out to the CZI team lead (Justin Kiggins) who is in charge of Starfish to see if we were missing anything (<https://forum.image.sc/t/unrecognized-version-number-for-spacex-files-created-with-java/57570/7> - you need to scroll down until you see the memory discussion). In a follow-up personal meeting, he confirmed that it is not possible, but they plan to address it in future versions.

RS-FISH overcomes this limitation using an ImgLib2-based block-wise implementation supporting overlaps with neighboring blocks that we can execute in a distributed fashion using Apache Spark.

b. Processing a reconstructed 148 GB lightsheet image stack took 32 CPU hours (~1 hour on a modern workstation). In comparison, a complex wrapping software for distributing AIRLOCALIZE, specifically developed for the EASI-FISH project to run on the HHMI Janelia cluster, required significant development effort and took 156 CPU hours to finish the same task.

Same as above, was the comparison made with any other tool, apart from AIRLOCALIZE?

We think that this is currently not possible with reasonable effort because other software does not support processing large volumes. We were only able to make that specific comparison with the ad-hoc extension of AIRLOCALIZE because our senior author is at Janelia - the large-data version of AIRLOCALIZE was written specifically for the Janelia cluster and only runs there (which was a project by itself). The development effort for that large-data version was significant, which included Matlab containerization using Docker and Singularity, ad-hoc handling of overlapping blocks, intensity normalization across blocks, and we actually found bugs related to the pixel origin when running it ourselves. We want to make the point that such extensions are (a) complex and (b) error-prone, thus we are not able to do that for any given software. That's why we are convinced that the support of large volumes by RS-FISH is a significant contribution. We believe that large image acquisition such as published by EASI-FISH⁷ recently, will be more common and analysis methods that are equipped to handle this smFISH data are missing.

Does the tool provide any registration functions for spatiotemporal or spatial transcriptomics methods, where the dots are detected from multiple imaging rounds. Also, does the position accuracy have any effect on

⁷ Wang Y., M. Eddison, G. Fleishman, M. Weigert, S. Xu, *et al.*, 2021 EASI-FISH for thick tissue defines lateral hypothalamus spatio-molecular organization. Cell 184: 6361-6377.e24.

these datasets, as this metric (supplementary table 1. Euclidean distance) is less accurate compared to other tools.

It is a very important point. RS-FISH itself does not provide any registration capability and we do not intend to add that since it is outside the scope of the software. However, in the future, we plan to use RS-FISH as one of the point detection algorithms for our BigStitcher project (<https://www.nature.com/articles/s41592-019-0501-0>), which we have already used to robustly align spatial genomics and transcriptomics dataset.

The spatial transcriptomics & spatial genomics datasets we have worked with so far all show low noise levels, thus the localization error is not lower for these cases (Fig. 2ef).

Parameter tuning is user dependent and having a lot of parameters to tune is a limitation that should be noted. We tested multi-consensus RANSAC on some of our multi channel smFISH images, but no dots were detected. We were not sure how to optimize settings to achieve detection. The authors should provide more sample data and details in the tutorial on how to use these different options on different images. Details on installation on different OS for batch processing should also be provided.

We thank the reviewer for pointing out that we could improve our explanation of the parameters. For this, we added a simple schematic (Supplemental Figure 7) on the role of each parameter for spot detection as well as a pipeline schematic for the different steps. We also added detailed guidelines to select parameters for RS-FISH, which is admittedly a bit more challenging for the multi-consensus case (section “**8. Choosing the right parameters for RS-FISH**” of the supplementary material). We hope this helps users to find the best parameters for their datasets.

Regarding, “installation on different OS for batch processing should also be provided”, it is the same on every operating system: Install Fiji (<http://fiji.sc>), add update site, and use Plugins > Macro > Record, which later can be run in an automated fashion. We now added a more detailed explanation also in section “**8. Choosing the right parameters for RS-FISH**” of the supplementary material.

Decision Letter, first revision:

Dear Dr. Preibisch,

Thank you for your patience as we've prepared the guidelines for final submission of your Nature Methods manuscript, "RS-FISH: Precise, interactive, fast, and scalable FISH spot detection" (N METH-BC47378A). Please carefully follow the step-by-step instructions provided in the attached file, and add a response in each row of the table to indicate the changes that you have made. Please also check and comment on any additional marked-up edits we have proposed within the text. Ensuring that each point

is addressed will help to ensure that your revised manuscript can be swiftly handed over to our production team.

In recognition of the time and expertise our reviewers provide to Nature Methods's editorial process, we would like to formally acknowledge their contribution to the external peer review of your manuscript entitled "RS-FISH: Precise, interactive, fast, and scalable FISH spot detection". For those reviewers who give their assent, we will be publishing their names alongside the published article.

Cover suggestions

As you prepare your final files we encourage you to consider whether you have any images or illustrations that may be appropriate for use on the cover of Nature Methods.

Nature Methods has now transitioned to a unified Rights Collection system which will allow our Author Services team to quickly and easily collect the rights and permissions required to publish your work. Approximately 10 days after your paper is formally accepted, you will receive an email in providing you with a link to complete the grant of rights. If your paper is eligible for Open Access, our Author Services team will also be in touch regarding any additional information that may be required to arrange payment for your article.

<https://mts-nmeth.nature.com/cgi-bin/main.plex?el=A3M7BCF5A7bhH4J3A9ftdnr3HPvPWS2mWK2n09BwwnQZ>

** Please ensure you delete the link to your author homepage in this e-mail if you wish to forward it to your co-authors. **

Best regards,
Amy Younger

Editorial Assistant
Nature Methods
methods@us.nature.com

On behalf of
Rita Strack, Ph.D.
Senior Editor
Nature Methods

Reviewer #1:
Remarks to the Author:

The authors have satisfactorily addressed my concerns. I think the revisions have improved the manuscript, and I support publication in Nature Methods.

Reviewer #2:

Remarks to the Author:

Reviewer name: Maurice Lee

The authors have revised the manuscript and addressed the earlier concerns. I really like the new addition of Supp Fig 7 where there is a very nice diagram to show the workflow for RS-FISH!

I have one concern that I should have brought up earlier. Is the current RS-FISH workflow only able to detect spots around a certain size range? Is it able to detect spots of different sizes/radii?

I have a very minor concern. It is the new use of the phrase "currently the only tool" for high detection accuracy and low localization error in the abstract.

The main problem is that one of the tools assessed in "Sage, D., Pham, TA., Babcock, H. et al. Super-resolution fight club: assessment of 2D and 3D single-molecule localization microscopy software. Nat Methods 16, 387–395 (2019)" may also have comparable high detection accuracy and low localization error as well.

Many thanks to the authors for doing the additional work to address the reviewers' concerns!

Reviewer #3:

Remarks to the Author:

The authors have addressed all of my comments. The added clarification, benchmarking, discussion, and new features of the method have strengthened the manuscript. Congrats to the authors on developing an excellent tool that will be widely used by the scientific community.

Author Rebuttal, first revision:

We thank all reviewers for their positive assessment and extremely valuable input on the manuscript

Reviewer #1:

Remarks to the Author:

The authors have satisfactorily addressed my concerns. I think the revisions have improved the manuscript, and I support publication in Nature Methods.

Thanks a lot!

Reviewer #2:

Remarks to the Author:

Reviewer name: Maurice Lee

The authors have revised the manuscript and addressed the earlier concerns. I really like the new addition of Supp Fig 7 where there is a very nice diagram to show the workflow for RS-FISH!

Thanks, we also think that it is very helpful for users of the software.

I have one concern that I should have brought up earlier. Is the current RS-FISH workflow only able to detect spots around a certain size range? Is it able to detect spots of different sizes/radii?

Yes, it can. We made it more clear in the text. Most importantly, the detection settings of the Difference-of-Gaussian need to be adjusted to get the right set of seed points. We now state in the main text: "RS-FISH first generates a set of seed points by thresholding the Difference-of-Gaussian (DoG) filtered image to identify potential locations of diffraction-limited spots, whose parameters need to be adjusted to the average size (sigma) and intensity (threshold) of the spots."

The actual RS localization should also be adjusted, although RS-FISH is less sensitive to these settings. Using a too-small area will simply lead to fewer gradients than possible being used, while a too big area will have more gradients that the RANSAC will filter out. We added the following to the methods section: "Importantly, the radius should be adjusted to the respective acquisition settings so that approximately double the size of the central peak of the PSF is entirely covered to ensure that all gradients that point towards the center of each spot are included in the localization."

I have a very minor concern. It is the new use of the phrase "currently the only tool" for high detection accuracy and low localization error in the abstract.

The main problem is that one of the tools assessed in "Sage, D., Pham, TA., Babcock, H. et al. Super-resolution fight club: assessment of 2D and 3D single-molecule localization microscopy software. Nat Methods 16, 387–395 (2019)" may also have comparable high detection accuracy and low localization error as well.

We changed the abstract to "RS-FISH maintains high detection accuracy and low localization error across a wide range of signal-to-noise ratios ...".

Many thanks to the authors for doing the additional work to address the reviewers' concerns! Reviewer #3:

Remarks to the Author:

The authors have addressed all of my comments. The added clarification, benchmarking, discussion, and new features of the method have strengthened the manuscript. Congrats to the authors on developing an excellent tool that will be widely used by the scientific community.

Thanks so much!

Final Decision Letter:

Dear Stephan,

I am pleased to inform you that your Brief Communication, "RS-FISH: Precise, interactive, fast, and scalable FISH spot detection", has now been accepted for publication in Nature Methods. Your paper is tentatively scheduled for publication in our December print issue, and will be published online prior to that. The received and accepted dates will be Oct 18, 2021 and Sept 28, 2022. This note is intended to let you know what to expect from us over the next month or so, and to let you know where to address any further questions.

Over the next few weeks, your paper will be copyedited to ensure that it conforms to Nature Methods style. Once your paper is typeset, you will receive an email with a link to choose the appropriate publishing options for your paper and our Author Services team will be in touch regarding any additional information that may be required.

Your paper will now be copyedited to ensure that it conforms to Nature Methods style. Once proofs are generated, they will be sent to you electronically and you will be asked to send a corrected version within 24 hours. It is extremely important that you let us know now whether you will be difficult to contact over the next month. If this is the case, we ask that you send us the contact information (email, phone and fax) of someone who will be able to check the proofs and deal with any last-minute problems.

If, when you receive your proof, you cannot meet the deadline, please inform us at rjsproduction@springernature.com immediately.

Once your manuscript is typeset and you have completed the appropriate grant of rights, you will receive a link to your electronic proof via email with a request to make any corrections within 48 hours. If, when you receive your proof, you cannot meet this deadline, please inform us at rjsproduction@springernature.com immediately.

Once your paper has been scheduled for online publication, the Nature press office will be in touch to confirm the details.

Content is published online weekly on Mondays and Thursdays, and the embargo is set at 16:00 London time (GMT)/11:00 am US Eastern time (EST) on the day of publication. If you need to know the exact publication date or when the news embargo will be lifted, please contact our press office after you have submitted your proof corrections. Now is the time to inform your Public Relations or Press Office about your paper, as they might be interested in promoting its publication. This will allow them time to prepare an accurate and satisfactory press release. Include your manuscript tracking number NMETH-BC47378B and the name of the journal, which they will need when they contact our office.

About one week before your paper is published online, we shall be distributing a press release to news organizations worldwide, which may include details of your work. We are happy for your institution or funding agency to prepare its own press release, but it must mention the embargo date and Nature Methods. Our Press Office will contact you closer to the time of publication, but if you or your Press Office have any inquiries in the meantime, please contact press@nature.com.

If you are active on Twitter, please e-mail me your and your coauthors' Twitter handles so that we may tag you when the paper is published.

Please note that Nature Methods is a Transformative Journal (TJ). Authors may publish their research with us through the traditional subscription access route or make their paper immediately open access through payment of an article-processing charge (APC). Authors will not be required to make a final decision about access to their article until it has been accepted. Find out more about Transformative Journals

Authors may need to take specific actions to achieve compliance with funder and institutional open access mandates. If your research is supported by a funder that requires immediate open access (e.g.

according to Plan S principles) then you should select the gold OA route, and we will direct you to the compliant route where possible. For authors selecting the subscription publication route, the journal's standard licensing terms will need to be accepted, including self-archiving policies. Those licensing terms will supersede any other terms that the author or any third party may assert apply to any version of the manuscript.

To assist our authors in disseminating their research to the broader community, our SharedIt initiative provides you with a unique shareable link that will allow anyone (with or without a subscription) to read the published article. Recipients of the link with a subscription will also be able to download and print the PDF. As soon as your article is published, you will receive an automated email with your shareable link.

Please note that you and your coauthors may order reprints and single copies of the issue containing your article through Springer Nature Limited's reprint website, which is located at <http://www.nature.com/reprints/author-reprints.html>. If there are any questions about reprints please send an email to author-reprints@nature.com and someone will assist you.

Best regards,
Rita

Rita Strack, Ph.D.
Senior Editor
Nature Methods